# Detecting Out-Of-Distribution Data with Semi-Supervised Feature Networks

## Abstract

Anomalous and out-of-distribution (OOD) data present a significant challenge to the robustness of decisions taken by deep neural networks, with myriad real-world consequences. State-of-the-art OOD detection techniques use embeddings learned by large pre-trained transformers. We demonstrate that graph structures and topological properties can be leveraged to detect both far-OOD and near-OOD data reliably, simply by characterising each data point (image) as a network of related features (visual concepts). Furthermore, we facilitate human-in-the-loop machine learning by expressing this data to comprise high-level domain-specific concepts. We obtained *97.95% AUROC* on far-OOD and *98.79% AUROC* on near-OOD detection tasks based on the LSUN dataset (comparable to the performance of state-of-the-art techniques).

## 1 Introduction

Trustworthy machine learning systems must hand over decisions it is not confident about to human experts. Most machine learning pipelines operate on the assumption of a closed world. The test data is assumed to be drawn in an IID fashion from the same distribution as the training data. The difficulty of OOD detection relies primarily on how semantically close the outliers are to the inliers. Therefore, based on difficulty [Winkens et al. (2020)], the OOD detection task is split into the following.

1. **Near OOD** refers to semantic shifts in the data, such as (SVHN and MNIST). Generally, this is a more challenging problem to solve, and the AUROC hovers around 93 per cent for state-of-the-art methods [Fort et al. (2021)].

2. **Far OOD** is a covariate shift, which is less difficult to detect. The AUROC hovers around 99 per cent in the current state of the art [Fort et al. (2021)].

Common sense is a very an essential yet absent element of AI systems. This crucial ability to judge and understand everyday things amongst most humans is a non-trivial problem with machines [Xu et al. (2021)]. The absence of common sense prevents intelligent systems from understanding a changing world (distribution drift), behaving reasonably in unforeseen situations (such as OOD detection), and learning quickly from new experiences (i.e. prior information). Furthermore, it is hard to learn, encode and represent this information. This shared and undefined knowledge base in humans is known from extensive exposure open domain data - such as basic physical phenomena.

In this paper, we operate under the assumption that common sense can be learnt in patterns of occurrences, and this knowledge can be learnt in a domain-specific manner. Therefore, our strategy relies on creating a commonsense service that learns from experience, based on computational models that mimic child cognition towards scenes and reasoning.

**Intuition** Graphs provide a general language for describing and analysing entities with interactions between them. We want to use the rich relational structures among visual concepts in complex domains to represent commonsense concepts. Our hypothesis is this would lead to better OOD prediction while maintaining justifications humans can understand.

**Contributions** This work includes the following contributions

1. We propose a novel semi-supervised geometric-learning-based framework that operates on human-interpretable concepts. This relies on representing each data point (image) as a graph of visual features.

2. We demonstrate that our technique performs on par with state-of-the-art methods on near and far-OOD tasks based on the LSUN dataset.

## 2 BACKGROUND AND RELATED WORK

**Detecting Out Of Distribution (OOD)** points in a relatively lower dimensional space have been extensively used [Pimentel et al. (2014)] in experiments. Conventionally, these methods include density estimation, nearest neighbour-based algorithms, and clustering analysis. The density estimation approach uses probabilistic models to estimate the in-distribution density, while declaring a data point out of distribution if it is located in a low-density region. Clustering-based methods rely on distance measures between points to find out-of-distribution points (that are further away from the neighbourhood). The primary drawback of these methods has always been their inadequacy in working with high-dimensional data [Theis et al. (2015)], such as images.

**Issues** Over the last years, state-of-the-art results in the OOD detection task have been based on deep neural network-dependent approaches. For example, convolutional neural networks have been used to find bizarre scenes by Sabokrou et al. (2018). Furthermore, the techniques presented in Andrews et al. (2016) and Fort et al. (2021) depend on an amalgamation of transfer-learning and representation learning. In sensitive environments, such as clinical settings, generative adversarial networks have been used by Schlegl et al. (2017). The drawback of a technique that requires adding more layers to a neural network or modifying its layers is that pre-trained neural networks - is that neural networks can be overconfident about wrong decisions when employed in an out-of-distribution setting [Hendrycks and Gimpel (2016); Lakshminarayanan et al. (2017); Guo et al. (2017)]. On the other hand, using large pre-trained transformer networks does improve performance - but relies heavily on the assumption that the embeddings generated by them are infallible.

**Scene Graph Generation (SGG)** refers to the task of automatically mapping an image or a video into a semantic structural scene graph, [Zhu et al. (2022) requiring the accurate labelling of detected objects and their relational structures. Although this is a tricky task, the availability of extensive datasets, such as Visual Genome [Krishna et al. (2016)], and massive models, such as OSCAR [Li et al. (2020); Zhang et al. (2021)] and RelationFormer [Shit et al. (2022)], has shown impressive results. In our case, since we do not need the relational structures between objects to be human-readable, in the interest of reducing computational overhead - we favour object detection networks over scene graph generation networks.

**Commonsense knowledge graphs (CSKGs)** are gaining popularity [Ilievski et al. (2021); Guan et al. (2019)] as origins of background knowledge (domain-specific conceptual, syntactic information) that are conceptualised to help with downstream reasoning tasks such as question answering and planning. In our context, we intend to use these for OOD data detection. For this, we exploit the recent advances in geometric learning. These same methods allow graph neural networks to be used to predict molecule properties and social media conversation characteristics.

## 3 PROBLEM SETUP

This paper assesses the problem of differentiating between in-distribution and out-of-distribution image examples on a pre-trained neural network. Let us assume that two distributions, $D_{in}$ and $D_{out}$, are drawn from the space $\mathcal{X}$. In dataset $\mathcal{D}^{\text{in}}$ of $\left(x^{\text{in}}, y^{\text{in}}\right)$ pairs where $x$ denotes the input feature vector, and $y^{\text{in}} \in \mathcal{Y}^{\text{in}} := \{1, \ldots, K\}$ denotes the class label. Let $\mathcal{D}^{out}$ denote an out-of-distribution dataset of $(x^{\text{out}}, y^{\text{out}})$ pairs where $y^{out} \in \mathcal{Y}^{\text{out}} := \{K+1, \ldots, K+O\}, \mathcal{Y}^{out} \cap \mathcal{Y}^{in} = \emptyset$. In our experiments, we sample from the mixture distribution. The conditional probability of drawing from this mixed distribution is $\mathcal{P}_{X|Z=0} = D_{in}$ for in-distribution and $\mathcal{P}_{X|Z=1} = D_{out}$ for out-of-distribution. Our problem setup allows access to OOD samples for training.

We are therefore presented with the following challenge: Given an image X drawn from the mixture distribution $P_{X*Z}$ - can we distinguish whether the image is from in-distribution $\mathcal{D}_{in}$?

## 4 USING FEATURE NETWORKS AS AN OUT-OF-DISTRIBUTION DETECTOR

This section presents our method for detecting out-of-distribution samples. This detector depends on a three-stage pipeline, the feature finder, the unsupervised projector, and the supervised detector. We describe the details of all three components below.

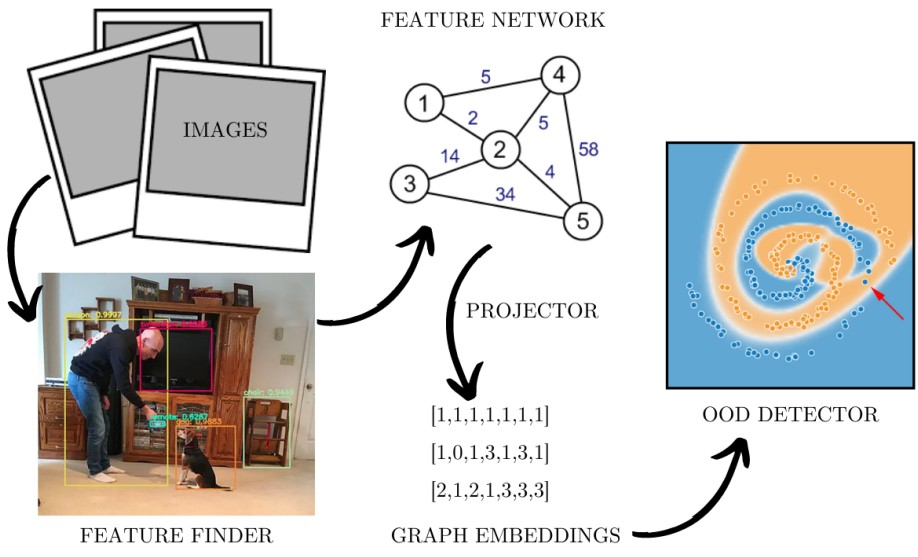

Figure 1: Stages for using Semi-Supervised Feature Networks for OOD detection.

### 4.1 FEATURE FINDER ($\phi : x_{input} \rightarrow \mathbb{G}$ )

We assume the availability of a pre-trained object-detection network that discovers the distinct components inside an image, draws boxes around them and assigns classes. Next, the proposed objects are pruned by thresholding based on the model's confidence (in section 5.4, we assume $\epsilon$=0.2)in the prediction. Finally, the stage generates the graph based on either of the following strategies :

- **Unweighted Graph** : Each class forms a node in the network, but edges are only drawn between the nodes found in the image. If multiple object pairs are found, multiple edges are drawn.

- **Weighted Graph** : As before, each class forms a node in the network, with edges only drawn between the nodes found in the image. If multiple object pairs are found, multiple edges are drawn - with pairwise weight factor assigned to each edge as a product of the intersection over union (IoU) score (Jaccard Index (J)) and the euclidean distance between the centroids of the two bounding boxes.

$$G(x) = (V, E); \mathbf{G}(x) \in \mathbb{G}; E_{obj_1,obj_2}^{weight} = \begin{cases} 1 + \|obj_1, obj_2\| * J(obj_1, obj_2) & \text{if weighted strategy} \\ 1 & \text{if unweighted strategy} \end{cases}$$

### 4.2 PROJECTOR ($\mathcal{P} : \mathbb{G} \rightarrow \mathbb{Z}$)

The previous stage's feature networks (graphs) represent each data point (image). Although the graph substructures, such as nodes and sub-graphs, are essential for our task - the best depiction of the entire data point and its intricacies can only be captured by the whole graph. Therefore, this stage focuses on creating a mapping from the structural and topological properties of this graph to a vector space which can be used for downstream analysis. Therefore this method performs whole-graph embedding, creating a projection into our "graph-feature" space.

Table 1: OOD detection tasks based on Yu et al. (2015)

| Far-OOD Tasks | Near-OOD Tasks |
|---|---|
| Bridge vs (Classroom, Conference Room, Dining Room, Kitchen, Living Room, Restaurant, Bedroom) | Bedroom vs (Classroom, Conference Room, Dining Room, Kitchen, Living Room, Restaurant); Living Room vs (Restaurant) |
| **Church Outdoor** vs (Classroom, Conference Room, **Dining Room**, Kitchen, Living Room, Restaurant, Bedroom) | Church Outdoor vs (Tower, Bridge), Bridge vs (Tower), Classroom vs (Conference Room, Dining Room, Kitchen, Living room, Restaurant); Kitchen vs (Living Room, Restaurant) |
| Tower vs (Classroom, Conference Room, Dining Room, Kitchen, Living Room, Restaurant, Bedroom) | Conference Room vs (Dining Room, Kitchen, Living Room, Restaurant); **Dining Room** vs (Kitchen, Restaurant, **Living Room**); |

Graph-kernel-based methods to perform whole-graph embeddings use handcrafted features such as shortest paths, graphlets etc. This hampers their performance and introduces problems such as poor generalisation. This is why, in this stage - we favour the usage of neural embedding frameworks that learn representations ($\mathbb{Z}$) and can scale to arbitrary-sized graphs.

### 4.3 DETECTOR ($\theta_z : \mathbb{Z} \rightarrow \hat{y}$)

Given the embeddings ($\mathbb{Z}$) learnt from the Graph Neural Network ($\mathcal{P}$), we use a downstream model to separate and bind the two distributions away from each other. Since the embeddings no longer have very high dimensions, we can leverage techniques from the rich literature on anomalous data detection. A classifier, $\theta_z$, may be trained to output whether the data point expressed by $Z_i$ is out of distribution or not ($\hat{y}$). If learning in an unsupervised manner, this includes density estimation, nearest neighbour, and clustering techniques.

## 5 EXPERIMENTS

### 5.1 OUT-OF-DISTRIBUTION DATASETS

**The Large scale Understanding Dataset (LSUN)** has about 120k to 3Mn images of 10 categories (each) such as bedroom, kitchen room, living room etc [Yu et al. (2015)]. We consider each of these categories as its domain. We classify pairwise combinations into far-OOD and near-OOD tasks depending on the semantic similarity between these domains. Out of the total 45 permutations possible (in table 1), we define pairs as near-OOD when both are outdoors or indoor scenes.

### 5.2 GRAPH EMBEDDING ALGORITHMS

1. **Graph2Vec** [Narayanan et al. (2017)] creates Weisfeiler-Lehman tree features for the nodes, based on which a feature co-occurrence matrix is developed and analysed to generate representations of the graphs. With a minimum count of feature occurrences as 5, we run this for 2 Weisfeiler-Lehman iterations, for ten epochs, with a learning rate of 0.025, and downsampling with frequency $10^{-4}$. This allows us to generate a 128-dimensional embedding vector.

2. **Wavelet Characteristic** [Wang et al. (2021)] uses characteristic functions of node features with wavelet function weights to describe node neighbourhoods. Once generated, node-level features are combined to create embeddings on the entire graph. We use averaging as the pooling function and run our experiments with a $\tau = 1.0$, $\theta_{max}=2.5$, 5 characteristic function evaluations and adjacency matrix powers set to 5. Finally, we generate a 1000-dimensional embedding.

3. **LDP** [Cai and Wang (2018)] relies on calculating the histograms of degree profiles, which are joined together to form graph representations. We run our experiments assuming 32 histogram bins and generate a 160-dimensional embedding.

4. **Feather Graph** [Rozemberczki and Sarkar (2020)] method utilises characteristic functions of node features with random walk weights to describe node neighbourhoods, which are combined to generate the entire graph's embeddings. We use averaging as the pooling function and run our experiments with $\theta_{max}$=2.5, 25 evaluation points and adjacency matrix powers set to 5. This generates a 500-dimensional embedding.

5. **Invariant Graph Embeddings** [Galland and Lelarge (2019)] computes a graph-descriptor using a mixture of spectral and node embedding-based features, such as eigenvalues and scattering. We use [10,20] as the number of histogram bins, [10,20] as spectral embedding dimensions, with [3,5] as feature embedding dimensions. This lets us generate a 220-dimensional embedding.

6. **GL2Vec** [Chen and Koga (2019)] generates embeddings while leveraging line graphs and edge features. It relies on decomposing a feature co-occurrence matrix of Weisfeiler-Lehman tree features for the nodes. With a minimum count of feature occurrences as 5, we run this for 2 Weisfeiler-Lehman iterations, for ten epochs, with a learning rate of 0.025, and downsampling with frequency $10^{-4}$. This allows us to generate a 128-dimensional embedding vector.

7. **NetLSD** [Tsitsulin et al. (2018)] generates 250-dimensional embeddings using the heat kernel trace of the normalised Laplacian matrix over time scales. We use 200 eigenvalue approximations, with 250-time scale steps and a time scale interval range between the minimum and maximum of -2.0 and 2.0, respectively.

8. **SF** [de Lara and Pineau (2018)] was used to generate 128-dimensional embeddings based on the lowest eigenvalues of the normalised laplacian.

9. **FGSD** [Verma and Zhang (2017)] was used to generate 200-dimensional embedding using the histogram of Moore-Penrose spectral features of the normalised laplacian. Here, we assumed 200 histogram bins, with a histogram range of 20.

## 5.3 Evaluation Metrics

We use these metrics to gauge the usefulness of our system in differentiating in and out of distribution images. Again, these are consistent with what's used to evaluate performance in OOD detection literature. [Fort et al. (2021); Liang et al. (2017)

1. **Area Under the Receiver Operating Characteristic curve** (AUROC) is a threshold-independent metric representing the relationship between TPR and FPR [Davis and Goadrich (2006)], where a perfect detector reaches an AUROC score of 100%. This can be thought of as the probability that an anomalous example is given a higher OOD score than an in-distribution example [Fawcett (2006)].

2. **AUPR** is another threshold-independent metric, [Manning and Schutze (1999); Saito and Rehmsmeier (2015)] evaluating the area under the PR (graph showing the precision=TP/(TP+FP) and recall=TP/(TP+FN), plotted against each other). It is useful when anomalous examples are infrequent, as it considers the base rate of the anomalies. We report the AP (average precision) score summarizes a precision-recall curve as the weighted mean of precisions achieved at each threshold, with the increase in recall from the previous threshold used as the weight.

## 5.4 Results

In this section, we demonstrate our method's effectiveness in detecting OOD datapoints on the LSUN-based benchmark (in table 1). We use an object detection model trained on Open Images V4 with ImageNet pre-trained Inception Resnet V2 as an image feature extractor to extract features. This specific model [Google (2022)] attains an mAP of 0.58 on the OpenImagesv4 test set and can successfully find 600 different classes.

Due to computational constraints, we test 20,000 images in each in-distribution and out-of-distribution classes on each near-OOD and far-OOD benchmark. In addition, 20% of the data is held out for unseen testing. The task for each category was randomly chosen from Table 1. In table 2, we describe the AUROC and average precision scores computed for Far-OOD detection between "Church (Outdoor)" and "Dining room" classes. Table 3 describes the AUROC and average precision scores computed for Near-OOD detection between "Living Room" and "Dining room" classes. For both these tasks, logistic regression and gradient boosting were used as $\theta_z$.

| Projection Method | AUROC | | | | AUPR (AP) | | | |
| | Logistic Reg | | Grad Boost | | Logistic Reg | | GradBoost | |
| (Accuracy) | *Test* | *Train* | *Test* | *Train* | *Test* | *Train* | *Test* | *Train* |
|---|---|---|---|---|---|---|---|---|
| **Wavelet Characteristic** | 84.65 | 84.77 | 85.50 | 85.58 | 82.72 | 82.88 | 82.75 | 82.84 |
| **LDP** | 85.51 | 85.58 | 85.50 | 85.58 | 82.82 | 82.86 | 82.75 | 82.84 |
| **FeatherGraph** | 84.81 | 84.99 | 85.50 | 85.58 | 82.81 | 83.06 | 82.75 | 82.84 |
| **IGE** | 83.66 | 83.11 | 85.44 | 85.77 | 82.54 | 82.40 | 83.27 | 83.48 |
| **GL2Vec** | 84.20 | 84.30 | 85.82 | 87.04 | 82.33 | 82.75 | 84.99 | 86.40 |
| **NetLSD** | 85.61 | 85.58 | 85.50 | 85.58 | 83.96 | 83.94 | 82.75 | 82.84 |
| **SF** | 85.45 | 85.54 | 85.50 | 85.58 | 83.33 | 83.54 | 82.75 | 82.84 |
| **FGSD** | 84.80 | 85.01 | 92.84 | 93.13 | 84.20 | 84.49 | 92.67 | 92.96 |
| **Graph2Vec (weighted)** | **94.47** | **94.25** | **97.95** | **98.22** | **94.34** | **94.15** | **97.69** | **98.14** |

Table 2: Results for Far-OOD detection were computed on the classes "Church Outdoor" and "Dining Room" based on table 1, from the LSUN dataset.

| Projection Method | AUROC | | | | AUPR (AP) | | | |
| | Logistic Reg | | Grad Boost | | Logistic Reg | | GradBoost | |
| (Accuracy) | *Test* | *Train* | *Test* | *Train* | *Test* | *Train* | *Test* | *Train* |
|---|---|---|---|---|---|---|---|---|
| **Wavelet Characteristic** | 73.20 | 73.63 | 72.99 | 73.51 | 72.33 | 72.27 | 70.54 | 70.57 |
| **LDP** | 72.99 | 73.50 | 72.99 | 73.51 | 70.56 | 70.53 | 70.54 | 70.57 |
| **FeatherGraph** | 73.29 | 73.78 | 72.99 | 73.51 | 71.73 | 71.78 | 70.54 | 70.57 |
| **IGE** | – | – | – | – | – | – | – | – |
| **GL2Vec** | 72.64 | 73.01 | 81.08 | 82.92 | 72.56 | 72.27 | 81.05 | 82.42 |
| **NetLSD** | 72.99 | 73.54 | 72.99 | 73.52 | 72.06 | 72.28 | 70.55 | 70.58 |
| **SF** | 73.06 | 73.57 | 72.99 | 73.51 | 72.05 | 72.06 | 70.54 | 70.57 |
| **FGSD** | 73.41 | 74.13 | 75.97 | 77.03 | 72.95 | 73.17 | 74.16 | 74.76 |
| **Graph2Vec (weighted)** | **93.51** | **93.46** | **98.48** | **98.79** | **92.73** | **92.73** | **98.53** | **98.78** |

Table 3: Results for Near-OOD detection were computed on the classes "Living Room" and "Dining Room" based on table 1, from the LSUN dataset.

# 6 DISCUSSION

In Fort et al. (2021) i.e. the current SOTA, an AUROC of 96% has been achieved in near-OOD detection, while an AUROC of 99% has been achieved in far-OOD detection. Considering that we achieved 98.79% AUROC on the near-OOD tasks from the LSUN dataset, and 97.95% AUROC on far-OOD tasks from the LSUN dataset - our model performance is comparable to the state-of-the-art.

Benchmarking and comparing the same tests are difficult since many image classification frameworks (CIFAR-10, 100) contain data points that contain only one feature variable (cars, planes), thereby negating the potency that graphs bring to express complex relationships inside data. Papers such as Fort et al. (2021) combine all the classes to create a distribution, such as all digits inside MNIST becoming one distribution. In extension, all digits inside SVHN become another distribution. Therefore, each of our classes inside the LSUN dataset is analogous to an entire distribution and not individual classes, such as cars, buses etc.

Just like words make up a document, the concept of visual words making up images has been studied in great detail. Topic models such as pLSA [Hofmann (2013)] worked on modelling co-occurence information under a probabilistic framework to discover underlying semantic structures.

**Handcrafted Features** Our semi-supervised featured networks are generated based on handcrafted features, such as human interpretable descriptors, euclidean distance and the jaccard Index. We have shown performance equivalent to state-of-the-art methods in our domain. However, in other domains good feature extraction may not be feasible, or an adequately large dataset may not be available. In future work, we plan to explore how these graphs can be built using lesser hand-engineering so that we can achieve better generalisation ability.

**Constraints** Our framework only requires access to an object detector that can find features of particular interest inside the required dataset and domain. It would also be helpful if this detector could find features that are expected to be seen in out-of-distribution data. Access to the list of features that can be detected could mean that the graphs in $\mathbb{G}$ contain nodes for every feature and only draw edges when pairwise relationships are observed., as described in section 4.1.

Using auxiliary models for sanity-checking the model's decisions is conceptually the same as boosting. This is because we effectively fit successive models to the residuals of previous models. We believe such a setup can reduce the risk of failure in production settings.

We observe that large amounts of graph data are essential for the self-supervised graph embedding algorithms to learn discriminative features maximally. When the same experiments were performed with 1000 images each, performance was generally not outstanding. This is possible because when the embedding $\mathbb{Z}$ is in high enough dimensions, and there are not enough data points -$\theta_z$ can always fit the training set, but generalisation on the test set is inferior.

A fundamental assumption in our framework currently is access to both the in and out-of-distribution data. It should be possible to extend our method not to require access to OOD samples, whether to train auxiliary models or to tune hyperparameters. However, if P has not trained on a varied amount of data - the discriminative ability may be limited, thereby compromising the performance of the OOD detector. In future experiments, we intend to examine methods to extend our framework, including the ability for zero or few-shot OOD detection.

## 7    CONCLUSION

Deep neural networks can make wrong decisions with high confidence when operating on data it was not trained on. Therefore, for the safe deployment of AI systems in the real world, model performance and detection of out-of-distribution data are equally important. To tackle this problem and find OOD data - we propose a novel semi-supervised geometric-learning-based framework that operates on human-interpretable concepts. We demonstrate that our technique performs on par with state-of-the-art methods on near and far-OOD tasks. In future work, we intend to explore more unsupervised methods to generate feature graphs and test our method on few-shot OOD detection.

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
