# OpenReview forum: "Detecting Out-of-Distribution Data with Semi-supervised Graph “Feature" Networks"
_ICLR.cc/2023/Conference — Submitted to ICLR 2023_

### Official Review · Reviewer_qAw3 · 2022-10-12

**Confidence:** 5
**Clarity, Quality, Novelty And Reproducibility:** 1. Experimental section is far from e…
**Correctness:** 3
**Technical Novelty And Significance:** 3
**Empirical Novelty And Significance:** 2
**Recommendation:** 3

**Strength And Weaknesses:**


Strength:

	1. The idea of building a semantic map to detect OOD is novel. It points out one promising direction for the interpretable OOD detection, which can tell "why the sample is an OOD".

	2. The comparison of different graph embedding algorithms is comprehensive.

Weakness:

	1. The proposed method relies on a pretrained object detection network that contains the sufficient semantic information for the in-distribution data. When the semantic of in-distribution data like medical images is not covered by the object detection network (pre-trained on natural image), the built semantic graph can be incomplete or even erroneous. If we use additional annotations to train a sufficiently strong object detection network, the effort will be extremely expensive comparing the existing methods.

	2. The paper is not polished and not ready to publish, with missing details in related work / experiment / writing. See more in "Clarity, Quality, Novelty And Reproducibility".


**Summary Of The Paper:**

This paper tackles the problem of Out-of-distribution detection. A pre-trained object-detection network is firstly used to build a semantic graph. Then the graph embedding is further used to detect OOD samples. The experiment is conducted on the far-OOD and near-OOD setting.

**Summary Of The Review:**

This paper provides an interesting idea that uses the pre-trained object detector to build semantic graph and then detect OOD samples by the graph embedding. Firstly, the assumption of having an oracle object detector for all in-distribution data is unrealistic. Secondly, the idea is poorly presented by the insufficient details and the experimental results are limited. Overall, the paper is far from ready to be published in this venue.

---

### Official Review · Reviewer_sju8 · 2022-10-23

**Confidence:** 3
**Correctness:** 2
**Technical Novelty And Significance:** 2
**Empirical Novelty And Significance:** 2
**Recommendation:** 3

**Clarity, Quality, Novelty And Reproducibility:**

This paper lack novelty and misses lots of detail. I believe the reproducibility of this paper is good, but it lacks detailed analysis of this method in this paper.

**Strength And Weaknesses:**

Strength:
1. The idea of this paper is interesting. I am not sure similar idea was proposed before or not.
2. This paper got results by using different graph embedding algorithms, it should be valuable for future work.
3. Based on the DISCUSSION part of this paper, the results of this should not be bad.

Weaknesses:
1. The method of this paper lack novelty. It combined several well-defined methods to do out-of-distribution detection.
2. I am not an expert in this domain, not sure if it is fine or not that this paper only shows their own results and didn't compare with other state-of-art methods.
3. I think this paper is not in good written. It lacks much detail about those methods, like what's the architecture of the detector. What if the pre-trained object detection network doesn't include the class for the out-of-distribution dataset?
4. This paper lack ablation studies like how the graph affects the results.

**Summary Of The Paper:**

This paper proposed a new pipeline that includes 3 stages for detecting out-of-distribution data. First, a graph is generated for an image by using its output from a pre-trained object-detection network. Secondly, the graph-kernel-based method is used to generate the whole-graph embedding from the graph. Finally, given the embedding from the graph neural network, a classifier is trained to detect the out-of-distribution data.

**Summary Of The Review:**

I think the idea of this paper is not novel enough and the experiments are not persuasive.

---

### Official Review · Reviewer_ox4H · 2022-11-02

**Confidence:** 5
**Correctness:** 1
**Technical Novelty And Significance:** 1
**Empirical Novelty And Significance:** 1
**Recommendation:** 1

**Clarity, Quality, Novelty And Reproducibility:**

- This work is not well-prepared, so I cannot give a judgement for its novelty.
- The paper writing is poor and should be improved as described above.
- The empirical evaluation is not in a proper setting and also not extensive.

**Strength And Weaknesses:**

Strength

This work is not well-prepared so that I cannot give points of strength.

Weakness

1. This paper writing should be improved. First, the notations in this work are confused, such as "Near OOD" and "Far OOD". Their notations in the introduction seems different from those in the Table 1. Second, the motivation and the proposed method are not clearly described, so I cannot give a judgement for its novelty. Third, this work totally ignores the literature of OOD detection, which is generally introduced in the survey [1].

2. The empirical results are not convincing. The proposed method is not compared to existing state-of-the-art methods for OOD detection, which can be found in [1]. Instead, the authors compare the proposed method to some graph embedding algorithms, which is confused for me. Moreover, they just conduct experiments on the LSUN dataset, which is not extensive.


[1] Yang, Jingkang, et al. "Generalized out-of-distribution detection: A survey." arXiv preprint arXiv:2110.11334 (2021).

**Summary Of The Paper:**

In this paper, the authors focus on the OOD detection task, where they consider both near OOD and Far OOD. To this end, they aim to exploit graph structures and topological properties to improve this task. Specifically, they propose a semi-supervised geometric-learning-based framework that operates on human-interpretable concepts. Finally, the authors conduct experiments on the LSUN dataset to verify the performance of the proposed method.

**Summary Of The Review:**

This work is not well-prepared and also not conducted in a way of scientific research. So, I recommend a strong reject.

---

### Official Review · Reviewer_cXz2 · 2022-11-03

**Confidence:** 3
**Correctness:** 3
**Technical Novelty And Significance:** 3
**Empirical Novelty And Significance:** 2
**Recommendation:** 5

**Clarity, Quality, Novelty And Reproducibility:**

The paper presents an interesting idea, however more clear representation of the critical details of the paper would be very helpful to the reader e.g. A more detailed description of the feature extraction procedure, including an example which guides the reader through image->object detection->graph creation steps using actual data would be very helpful. Figure. 1 in its current form looks representative but it is not clear if it qualifies as an actual example.

Novelty of the paper lies in deriving visual semantic structures from images. It employs existing feature extraction techniques to convert these structures into representations suitable for training OOD detection models.

The paper utilizes an open dataset for experimental results, but the algorithm has not been described in detail sufficient for reproduction of the results.

**Strength And Weaknesses:**

Strengths
-----------

The paper clearly describes the intuitions and designs a feature extraction technique based on these.
It performs extensive comparisons using a variety of projection techniques to derive embeddings from the graph representation.

Weaknesses
--------------
The paper uses weakly defined terms like "Common sense". Defining these concepts in a more rigorous way would help the reader.

The authors summarize their work in the introduction. However it is not straightforward to relate this summary to the actual implementation.
It would be helpful if they better establish this relationship with the rest of the paper

 e.g.
"... Commonsense service that learns from experience"
 By common-sense service are they alluding to their object detection models and relationship generator ?
Is this "experience" their training data or they saying that they are creating a continuous learning system, which they aren't in this particular paper.
"computational models that mimic child cognition"
A strong reference which establishes the relationship with child cognition is necessary here.
The idea is intuitive and the authors claims are plausible, however the standard of this conference requires more rigorous demonstration of such claims.

The paper utilizes the performance reported in a reference to claim it is close to SoTA techniques. Typically such claims are validated by reproduction of the SoTA techniques for controlled comparisons.

One of the critical assumptions here is the accuracy of the object detection model. Though the authors have clarified that embedding extraction methods are susceptible to generate incorrect representations for out-of-domain data, they haven't provided an analysis of how the object detectors accuracy impacts the current model. This is a critical detail as OOD data affects all models.

There are quite a few grammatical mistakes in the paper which make it difficult to understand.

**Summary Of The Paper:**

In this paper authors propose a mechanism for deriving low-dimensional representations suitable for effective use of established non-parametric and parametric out-of-distribution data detection methods. Specifically they utilize graphs which represent relationships among the objects detected in an image. They claim that this low-dimensional representation mechanism mimics human cognitive processes and is better suited to detect novelty.

**Summary Of The Review:**

The paper translates an intuitive representation mechanism into a feature extraction technique. It demonstrates that this technique is highly competitive. However there are a few changes which could further strengthen the paper and make it more accessible to the reader. These include
* more detailed description of the feature extraction procedure with a real example
* stronger experimental validation by comparing to other low dimension projection techniques which can be utilized with OOD detection algorithms
* more rigorous/well referenced and less hand-wavy descriptions of the motivation/intuition

---

### Public Comment · ~Benedek_Andras_Rozemberczki1 · 2022-11-05
**Attribution of library used for experiments**

It is reasonable to assume that the paper uses the Karate Club library for the experiments extensively, yet it is not cited. Please add the citation if it was used for the experiments:

```bibtex
@inproceedings{karateclub,
       title = {{Karate Club: An API Oriented Open-source Python Framework for Unsupervised Learning on Graphs}},
       author = {Benedek Rozemberczki and Oliver Kiss and Rik Sarkar},
       year = {2020},
       pages = {3125–3132},
       booktitle = {Proceedings of the 29th ACM International Conference on Information and Knowledge Management (CIKM '20)},
       organization = {ACM},
}
```

---

### Decision · Program_Chairs · 2023-01-20

**Decision:**

Reject

**Justification For Why Not Higher Score:**

The paper has various technical and writing flaws that do not meet the ICLR acceptance bar.

**Justification For Why Not Lower Score:**

NA

**Metareview: Summary, Strengths And Weaknesses:**

This paper proposes leveraging visual semantic structures for the OOD detection problem. The semantic graph is generated by using output from a pre-trained object-detection network. As recognized by multiple reviewers, the idea of exploiting graph structure is interesting and has novelty.

However, reviewers identify several common weaknesses in the paper, including (1) the algorithm description lacks clarity and sufficient details, (2) the empirical evaluations are unconvincing, due to missing comparison with existing state-of-the-art methods for OOD detection.

All four reviewers unanimously voted rejection. The authors did not provide an author response in the rebuttal phase either.

The paper in its current form does not meet the ICLR standard and is therefore rejected.